# Evaluation of Fairness of Urban Park Green Space Based on an Improved Supply Model of Green Space: A Case Study of Beijing Central City

Xinke Wang [1,2], Qingyan Meng [2,3,4,*], Xingzhao Liu [1,5], Mona Allam [2,6], Linlin Zhang [2,3,4], Xinli Hu [2,3,4], Yaxin Bi [7] and Tamás Jancsóg [8]

1  College Landscape Architecture, Fujian Agriculture and Forestry University, Fuzhou 350002, China
2  Aerospace Information Research Institute, Chinese Academy of Sciences, Beijing 100094, China
3  University of Chinese Academy of Sciences, Beijing 100049, China
4  Key Laboratory of Earth Observation of Hainan Province, Hainan Aerospace Information Research Institute, Sanya 572029, China
5  Innovation Center of Engineering Technology for Monitoring and Restoration of Ecological Fragile Areas in Southeast China, Ministry of Natural Resources, Fuzhou 350013, China
6  Environment & Climate Changes Research Institute, National Water Research Center, El Qanater EI Khairiya 13621/5, Egypt
7  School of Computing, Ulster University, York Street, Belfast BT15 1ED, UK
8  Alba Regia Technical Faculty, Óbuda University, Budai ut 45, H-8000 Szekesfehervar, Hungary
*  Correspondence: mengqy@radi.ac.cn

**Abstract:** Urban park green space (UPGS) plays an important role in providing ecological and social benefits. However, in many large cities with rapid economic development, the supply of UPGS is unfairly distributed, and there is a severe mismatch between its supply and residents' demand. Taking the Beijing central city as an example, this study aims to develop a fairness assessment model to quantify the fairness of UPGS distribution and the matching relationships between supply and demand for UPGS. To achieve the aims of the study, we improved the supply model of UPGS by integrating three factors: the number of UPGS, the service capacity of UPGS, and the quality of UPGS in the Beijing central city. Subsequently, we evaluated the spatial fairness and social fairness of the supply of UPGS using the Gini coefficient. Then, we used the number of residents in the sub-district to characterize the intensity of residents' needs and quantitatively analyzed the spatial matching relationship between the supply of UPGS and residents' demand. The results show that: (a) The improved supply model of UPGS can measure the supply of UPGS of different types in a more detailed way. (b) The per capita supply of UPGS is unevenly distributed among the six urban districts of Beijing, which may lead to a sense of unfairness among residents. While residents in Haidian District (Gini = 0.649) may have the highest sense of unfairness, followed by those in Fengtai (Gini = 0.505), Dongcheng (Gini = 0.410), Xicheng (Gini = 0.392), and Chaoyang District (Gini = 0.225). (c) The matching relationship between the supply of UPGS and the needs of different social groups is not ideal, especially the spatial matching relationship between the needs of the elderly and the supply of UPGS. This study can be used as a reference for supporting decision making in optimizing UPGS and providing a reference for fine urban management.

**Keywords:** urban park green space; supply model of UPGS; spatial fairness; social fairness; matching supply and demand

## 1. Introduction

Fairness and justice are an important symbol of human civilization and a standard for measuring the civilized development of a country or society. In 1965, Adams proposed the theory of perceived fairness. That is, in social relationships, people always compare what they get in a social exchange with others and make judgments of fairness or unfairness,

thus creating a sense of fairness [1]. Fairness is the residents' evaluation of the fairness of opportunity, process, and distribution in daily life, and is the touchstone to test the harmony of social livelihood [2]. In 2017, the basic contradiction of Chinese society changed to "the contradiction between the people's growing need for a better life and unbalanced and insufficient development." Imbalances are issues of fairness, including uneven development between regions, social divisions between groups, and mismatches between physical space and social groups. Rapid economic growth has resulted in increasing social stratification and the division of living space in recent years, and people cannot enjoy public resources fairly, particularly urban park green space (UPGS) [3–6]. UPGS can improve the living environment and play a vital role in cities [7], such as alleviating the heat island effect, regulating microclimates, containing water, reducing noise [8], treating diseases, etc. [9]. However, the supply of UPGS is gradually failing to meet the demand of urban residents, and the problem of the unfairness of UPGS is becoming more and more pronounced [10]. The needs of residents referred to in this study are the need for access to UPGS. In general, socioeconomic data and service beneficiary perception survey data are commonly used to evaluate demand indicators [11,12]. With the development of networks and big data, social media data [13], cell phone signaling data [14], etc., are also gradually being used to reflect the real demands of residents for recreational space. However, due to the availability of data, some scholars are still using socio-economic demographic grid data, gross domestic product, and nighttime lighting data to indicate residents' demand [15]. In this study, the number of residents in a sub-district is used to indicate potential residents' demand. As an important way to improve the urban environment and residents' quality of life, the construction of UPGS bears the responsibility of building social fairness [16]. Therefore, China urgently needs to construct a system for evaluating the fairness of UPGS to cope with the current contradiction between the growing demand for residents' living and the unbalanced development of UPGS.

There are already many scholars who are concerned about the fairness of UPGS. Research on the fairness of UPGS can be summarized in three stages: spatial fairness, social fairness, and social justice. Spatial fairness refers to the fair distribution of the supply of UPGS. It is an idealized state that does not consider the needs of urban residents. Currently, scholars mostly use the Gini coefficient and Lorenz curve to evaluate spatial fairness [3,17]. The Gini coefficient and the Lorenz curve are commonly used in economics to measure the income disparity between residents of a country or region. With the development of cross-fertilization of disciplines, the Gini coefficient is gradually used to evaluate the fairness of the spatial distribution of public resources such as education [18] and UPGS [19]. Social fairness focuses on the "spatial match" between the distribution of green space resources and the residential population [2]. Social justice is based on social fairness and emphasizes that the capabilities and needs of various groups are different. It believes that different socio-economic groups should have access to public resources fairly and advocates that disadvantaged groups should have greater access to public resources [20–22].

Indicator metrics are the basis of evaluation. Scholars often use indicators such as green space per capita [23], the number of parks within a certain distance [24], and the accessibility of parkland [25,26] to measure the supply of UPGS, which are still in a single dimension and do not fully reflect the characteristics of UPGS. At present, scholars mostly rely on field questionnaires to assess UPGS's quality [27], which are labor-intensive and costly, and less often consider the quality of UPGS from the perspective of big data. He et al. constructed a supply metric model of UPGS by integrating UPGS's quality and UPGS's service capacity [28]. However, the UPGS's quality index in this model is not quantified, and the assessment method is not precise. Moreover, the service capacity of UPGS of different types is not distinguished. Therefore, this study improved the supply model of UPGS proposed by He et al. In recent years, big data have provided new methods for social science practice [29]. In this study, we obtained the quality factor data of UPGS from big datasets such as AOI, POI, and OSM. We then introduced a new data source, the Baidu heat map, to represent the visits of UPGS, making it the basis for establishing the weights of the

quality factors of UPGS. This method weakened the previous subjectivity in establishing the quality factors of UPGS. In addition, this study determined the service radius of UPGS based on their scale and measured the supply of UPGS of different types, as well as having improved the supply model of UPGS to guide optimum urban management.

Beijing, as the capital of China, has experienced rapid economic development and an increasingly apparent trend of imbalance between the supply and demand of public services [30,31]. Therefore, this study selected Beijing as a case study and integrated the quantity of UPGS, UPGS's service capacity, and UPGS's quality to measure the supply of UPGS. Subsequently, the Gini coefficient was applied to evaluate the fairness of the supply of UPGS from spatial and social fairness perspectives. This study analyzed the match relationship between supply of UPGS and residents' demand in the central urban district of Beijing and identified the areas with insufficient supply of UPGS. The empirical study of fairness in UPGS in China can consolidate the relevant theoretical framework of green space fairness and provide comparative results for existing studies [32–34]. Moreover, especially for different social groups, it is the basis for maximizing social benefits for local governments [28].

The overall objectives of this study are: (a) to improve the metric model of supply of UPGS proposed by He et al. by combining the number of UPGS, UPGS's service capacity, and UPGS's quality; (b) to explore the fairness of the supply of UPGS from the perspective of spatial fairness and social fairness; and (c) to explore the spatial match between the supply of UPGS and residents' demand (Figure 1).

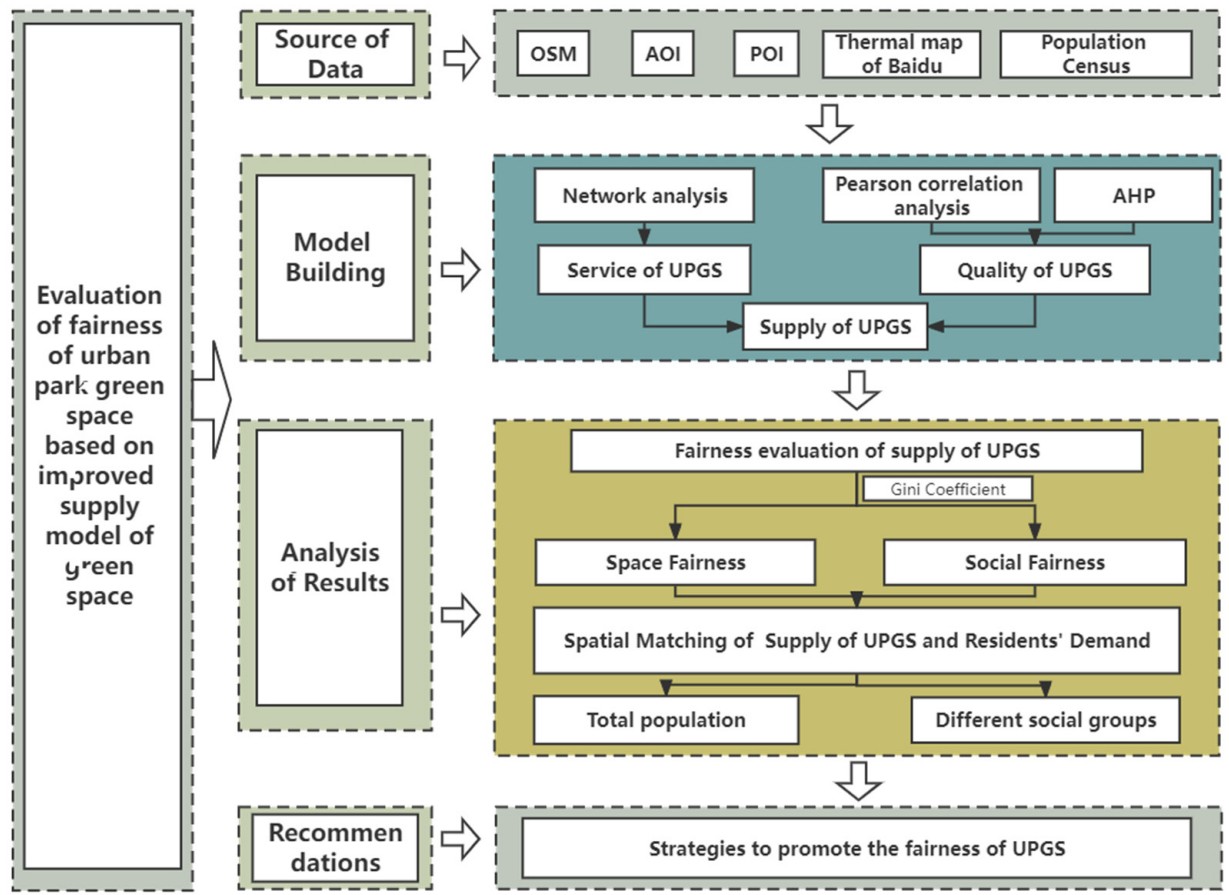

**Figure 1.** Technical route.

## 2. Materials

### 2.1. Study Area

Beijing, located at longitude 116°20′ East and latitude 39°56′ North, is China's political, cultural, and international exchange center (Figure 2). In recent years, Beijing's rapid economic development has resulted in a growing trend of imbalances between supply and demand for public services [30,31]. In 2020, Beijing had a resident population of 21,893,000, an increase of 2,281,000 compared to the resident population in 2010. It is calculated that the average annual growth of Beijing is 228,000 people, and the demand for residents has increased significantly. This study focuses on the central urban district of Beijing, including Chaoyang, Haidian, Dongcheng, Xicheng, Fengtai and Shijingshan District. This area covers less than 10% of the total area of Beijing, but the total number of residents accounts for more than 50% of the total population of Beijing [35].

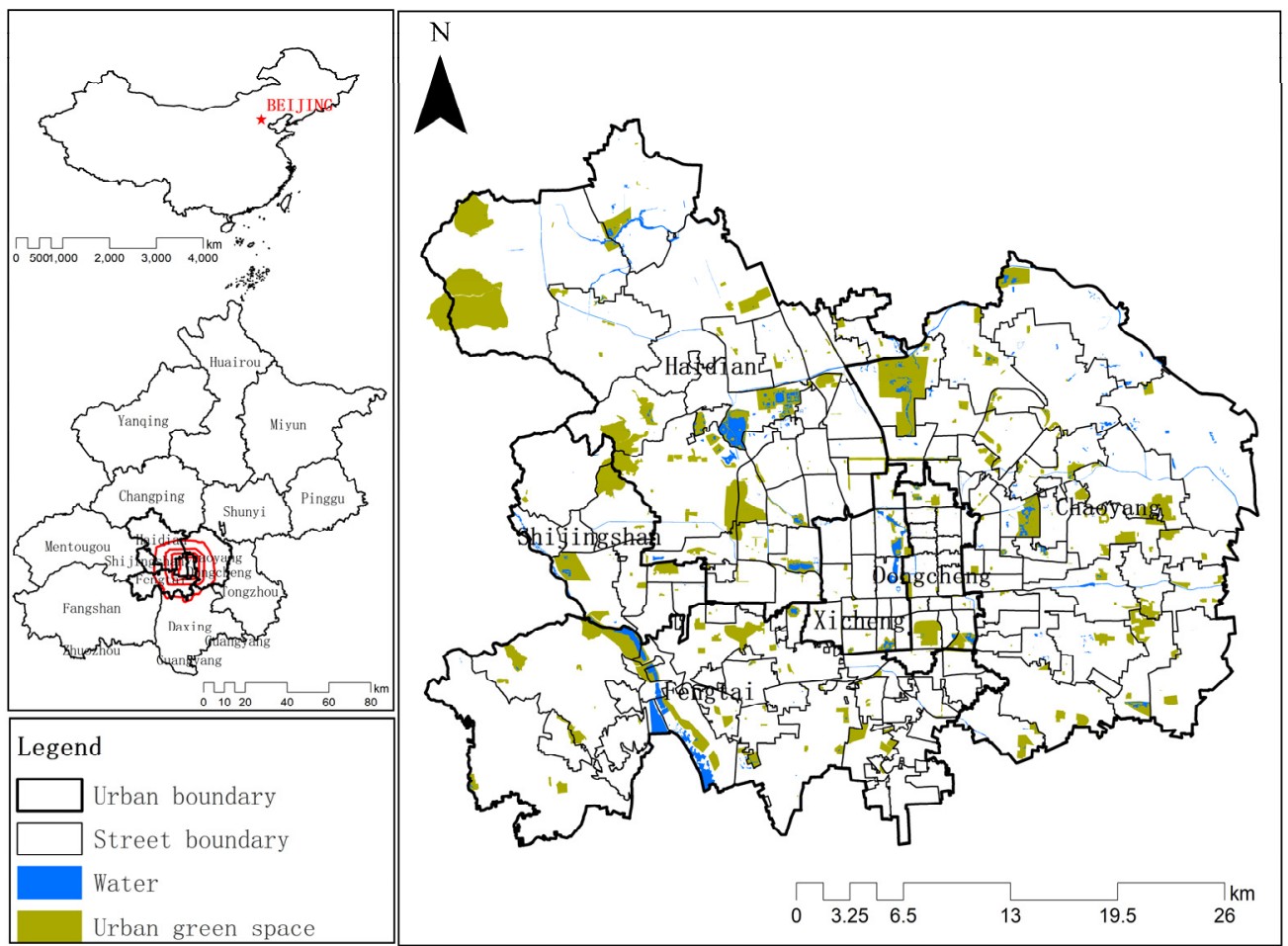

**Figure 2.** Study area.

According to survey data from the Beijing Municipal Bureau of Statistics, the Beijing Municipal Government has vigorously increased the city's green coverage, with the per capita park green space increasing from 5.1 m$^2$ in 1978 to 16.4 m$^2$ in 2019. However, we argue that green coverage and green space per capita only reflect the quantitative characteristics of UPGS without considering the quality of UPGS. Increasing green space per capita does not significantly improve public green space's social benefits [36,37]. Considering the quality of UPGS, UPGS's service capacity and other factors, an in-depth exploration of the supply of UPGS of different types in Beijing is essential for the refined management of compact cities such as Beijing.

*2.2. Data Description and Pre-Processing*

All geographic data used in this study are in the WGS 1984 UTM Zone 50N projection coordinate system.

2.2.1. The Vector Boundary of the Sub-district and Population Data within the Sub-district

The unit of analysis for this study was the sub-district, and the sub-district boundaries were obtained from the data published in the National Basic Geographic Information Database in July 2021 (http://www.ngcc.cn/ngcc, access on 17 October 2021). We topologically examined these data and finally extracted 133 sub-districts in the study area. Among them, 17 are in Dongcheng District, 15 in Xicheng District, 42 in Chaoyang District, 29 in Haidian District, 21 in Fengtai District, and 9 in Shijingshan District. According to China's three-level administrative division system, the sub-district level (similar to the U.S. census district level) is the smallest administrative unit [25].

The total population data for the sub-districts was counted from the 7th Census data in 2020, which were obtained from the Beijing Municipal Bureau of Statistics (http://tjj.beijing.gov.cn/, accessed on 1 October 2022). We obtained data on the elderly, youth, and female groups in the sub-districts by correcting the data from the 6th Census and the 7th Census announcement data.

2.2.2. Area of Interest (AOI)

AOI can provide an accurate geographical location and detailed attribute information for urban facilities. This study crawled the AOI data of Beijing using Python based on the Baidu map platform (https://lbsyun.baidu.com, accessed on 10 October 2021), and then extracted the scenic spot types in the AOI and corrected them by comparing them with the GF-1 satellite image data, after filtering, cleaning, and topology checking to eliminate the non-UPGS such as temples and pavilions, and finally obtained the vector boundaries of 340 parks and their attribute information. The attribute information includes the park name, longitude, latitude, and area.

2.2.3. Point of Interest (POI)

POI can also reflect the precise location of urban facilities. This study crawled POI data based on the Baidu Map platform (https://lbsyun.baidu.com, accessed on 10 October 2021), and subsequently selected the public service facility class. Finally, the number of public service facilities within all parks in Beijing was obtained after overlay analysis processing. These data were used as the input for the park quality factor.

2.2.4. Road Network Data

First, we downloaded the road data from the OSM website (https://www.openstreetmap.org/, accessed on 18 October 2021), including attributes such as road name, length, and grade. The OSM website is a publicly available street map. Next, we filtered five types of roads from the data, namely, primary, secondary, tertiary, trunk, and residential, and then checked them. Finally, we corrected them by referring to the "Beijing 13th Five-Year Plan" and combining the Baidu map and the Gaode map. In addition, the park's water and green space boundaries and their attributes were obtained from the OSM, which were used as inputs for the park quality factors.

2.2.5. Population Heat Value

A Baidu heat map is a kind of internet open-source data that can dynamically reflect the characteristics of urban population gathering [38]. Currently, it is often used to reflect urban spatial vitality [39], park accessibility [40], and the practical intensity characteristics of urban space [41]. Studies have shown that on rest days, people prefer to travel in the afternoon and evening, reaching the peak of the day around 15:00; on weekdays, population activity shows a clear "morning peak" and "evening peak" of commuting [42]. This study crawled the Baidu population heat data at 3:00 pm on 9 October 2020 (Friday),

10 October 2020 (Saturday), 16 October 2020 (Friday), 17 October 2020 (Saturday), 23 October 2020 (Friday), and 24 October 2020 (Saturday) based on the Baidu Maps platform (https://lbsyun.baidu.com, access on 9, 10, 16, 17, 23, and 24 October 2021). The original data of the Baidu heat map obtained in this study are a series of points spaced about 100 m apart. First, based on the ArcGIS 10.2 platform, we performed kernel density analysis on the Baidu thermal data at each of the six moments, setting the spatial resolution to 2 m. After that, based on the spatial masking tool in ArcGIS 10.2, we spatially masked the Baidu thermal map with the UPGS. Subsequently, using the zonal statistics tool, we calculated the total thermal values inside each UPGS for six moments separately. Finally, we calculated the average heat value within the UPGS on Friday and Saturday, respectively, representing the park visitation.

### 2.2.6. Park Quality Factor

It is widely accepted that high proximity to UPGS does not mean that it is sufficiently attractive to residents [43]. The park quality often plays a critical role in people's choice of UPGS and determines how long users stay. It is well-documented that park composition, spatial configuration, and other micro-features, such as the size of UPGS, presence or absence of water within the UPGS, walkable environment within the UPGS, and canopy cover within the UPGS, affect how often residents are exposed to the UPGS and how long they stay in the UPGS [44–46]. Therefore, in this study, the proportion of green space in the UPGS to the total park area, the proportion of water body area in the UPGS to the whole area of UPGS, the whole area of UPGS, and the number of public services in the UPGS were selected as the quality factors of UPGS. The four factors were finally graded according to the natural interruption method and reassigned. In this study, the proportion of the area of water bodies in the UPGS to the total area of the UPGS, and the proportion of the area of green space in the UPGS to the total area of the UPGS were selected for the evaluation of the quality of UPGS, eliminating the influence of scale of UPGS on the area of water bodies and the area of green space.

## 3. Methods

The methodology used in this study is divided into the following parts: (1) using Pearson correlation analysis and Analysis Hierarchical Process to improve the supply model of UPGS; (2) using the Gini coefficient to assess the fairness of UPGS; and (3) applying the supply and demand relationship to evaluate the spatial match between the supply of UPGS and residents' demand.

### 3.1. Improvement of Supply Metric Model of UPGS

This study improved the supply model of UPGS proposed by He et al. (He. et al., 2020). He et al. did not consider the classification of UPGS and did not specify a specific method for quantifying UPGS's quality when constructing the supply model for UPGS. Therefore, this study considered the service radius of UPGS of different types and the explicit quality measurement method of UPGS based on big data when constructing the supply model of UPGS, which makes the evaluation model more objective.

### 3.1.1. The Service Capacity of UPGS

The UPGS's service capacity is related to the scale of UPGS and the road network. This study calculated the service area for UPGS based on the service area analysis in the ArcGIS platform. According to the Urban Land Classification and Planning and Construction Land Standard (GB50137-2011), this study classified UPGS into small parks (S < 1 hm$^2$), community parks (1 hm$^2$–10 hm$^2$), regional parks (10 hm$^2$–25 hm$^2$), and city parks (S > 25 hm$^2$) (Table 1), where S is the park scale. As shown in Table 2, the service radius of small, community, regional and city park is 500 m, 1000 m, 1500 m, and 3000 m, respectively [47].

**Table 1.** Statistics on the classification of UPGS.

| Space Level | Number | Area (hm²) | Decision Criteria | Service Radius (m) |
|---|---|---|---|---|
| Small Park | 57 | 30.447 | $Si < 1\ hm^2$ | 500 |
| Community Park | 150 | 599.163 | $1\ hm^2 \leq Si < 10\ hm^2$ | 1000 |
| Regional Park | 48 | 788.371668 | $10\ hm^2 \leq Si < 25\ hm^2$ | 2000 |
| City Park | 85 | 10,214.442372 | $Si > 25\ hm^2$ | 3000 |

**Table 2.** The grading status of UPGS quality factors.

| Categories | Grading | Score |
|---|---|---|
| WB (Proportion of water body area in the UPGS to the area of the UPGS/%) | <2.95% | 1 |
| | 2.95%–14.27% | 3 |
| | 14.27%–32.96% | 5 |
| | 32.96%–62.73% | 7 |
| CC (Proportion of green space in the UPGS to the area of the UPGS/%) | <58.67% | 1 |
| | 58.67%–84.01% | 3 |
| | 84.01%–96.31% | 5 |
| | 96.31%–100% | 7 |
| GS (UPGS' area/m²) | <1174205.27 | 1 |
| | 1,174,205.27–2,845,955.21 | 3 |
| | 2,845,955.21–5,157,949.82 | 5 |
| | 5,157,949.82–9,070,556.09 | 7 |
| PF (Number of public service facilities in the UPGS/each) | <7 | 1 |
| | 7–21 | 3 |
| | 21–41 | 5 |
| | 41–117 | 7 |

### 3.1.2. Quality of UPGS

The quality of UPGS is calculated as Equation (1). Where a + b + c + d = 1. GQ is the quality score of UPGS. GS is the score of scale of UPGS; CC is the score of green area in the UPGS; WB is the score of water area in the UPGS; PS is the score of the number of public facilities in the UPGS. The factor classification is shown in Table 2. a, b, c, d are the weight coefficients of the four urban park quality factors, respectively.

In this study, Pearson correlation analysis was performed between the quality factor of urban green space and the number of visits to the UPGS. In this case, the correlation coefficient q of the two factors can be used as a basis for a two-by-two comparison of the importance of the factors in the Analytic Hierarchy Process (AHP).

$$GQ = a*GS + b*CC + c*WB + d*PF \tag{1}$$

(1)    Pearson correlation analysis

The Pearson correlation coefficient, also known as the Pearson product–moment correlation coefficient (PPMCC or PCCs), is a measure of the correlation (linear correlation) between two variables X and Y, and its value is between −1 and 1 [48].

(2)    Analytic Hierarchy Process

AHP is a decision-making method that decomposes the elements related to decision making into levels such as objectives, criteria, and solutions, based on which qualitative and quantitative analysis are performed [49]. The first step of AHP is to construct a hierarchical structure, which includes goal, criterion, and solution levels [50]. In the second step, a two-by-two priority comparison of the elements of each hierarchy level based on a priori knowledge is required, which can be conducted using Saaty's 1–9 scale method [51]. Finally, the two-by-two comparison matrix is tested using the consistency test Formulas (2) and (3). Where $\lambda_{max}$ is the maximum eigenvalue of the pairwise comparison matrix and n is the

order of the matrix, there is a correspondence between the random index (RI) and n. i.e., the RI is 0, 0, 0.58, 0.90, 1.12, 1.24, 1.32, 1.41, 1.45, and 1.49 for n from 1 to 10. If CR is 0.1, the weight distribution is reasonable [35].

$$CR = \frac{CI}{RI} \tag{2}$$

$$CI = \frac{\lambda_{max} - \mathbf{n}}{\mathbf{n} - 1} \tag{3}$$

*3.2. Gini Coefficient*

The Gini coefficient and the Lorenz curve are often used to measure the income disparity between residents in a country or region, and are now mostly used to measure the equilibrium of resource distribution. According to international standards, the Gini coefficient is relatively average between 0.2 and 0.3, relatively reasonable between 0.3 and 0.4, and there is a large gap between 0.4 and 0.5 [52]. The Gini index usually takes 0.4 as the "warning line" for the gap in resource allocation, and its exact value should be 0.382 according to the golden mean.

$$\mathbf{G} = \mathbf{1} - \frac{\mathbf{1}}{\mathbf{m}} \sum_{\mathbf{i=1}}^{\mathbf{m}} \frac{(\mathbf{C_i} + \mathbf{C_{i-1}})}{\sum_{\mathbf{i=1}}^{\mathbf{m}} \mathbf{C_i}} \tag{4}$$

where the total number of sub-districts is m; $\mathbf{C_i}$ is the ranked cumulative value of supply of UPGS. The calculation is as follows: the supply A of UPGS of m sub-district is sorted from smallest to largest and accumulated, which is the total sum of A1 to Ai.

*3.3. Matching Supply and Demand*

Jenks' best natural break method is a map grading algorithm proposed by Jenks. He believes that the data themselves have breakpoints and can be graded using this feature of the data. The classification principle of this method is that the variance between groups is as large as possible and the variance within groups is as small as possible [14,15]. In this study, we use Jenks' best natural break method in ArcGIS 10.2 to classify the supply of UPGS enjoyed by the sub-district as low supply, medium supply, and high supply, and the number of residents in the sub-district as low need, medium need, and high need, and assign them to the attribute table of ArcGIS 10.2 as 1, 3, and 5, respectively [14]. Then, we conduct algebraic operations on the supply of UPGS and the need for residents, as in Equation (5), and finally generate the supply–demand relationship table (Table 3).This study concluded that sub-district with low supply-low demand (Low–Low), medium supply-medium demand (Mid–Mid), and high supply-high demand (High–High) all reach supply–demand equilibrium; sub-districts with low supply-medium demand (Low–Mid), low supply-high demand (Low–High), and medium supply-high demand are under-supplied; and sub-district with medium supply-low demand (Mid–Low), high supply-low demand (High–Low), and high supply-medium demand (High–Mid) are over-supplied [14].

$$SD = Supply*100 + Demand \tag{5}$$

**Table 3.** Supply and demand matching relationship.

| Matching Supply and Demand | Low Supply (1) | Medium Supply (3) | High Supply (5) |
|---|---|---|---|
| **Low demand (1)** | Low supply - Low demand (101) | Medium supply - Low demand (301) | High supply - Low demand (305) |
| **Medium demand (3)** | Low supply - Medium demand (103) | Medium supply - Medium Demand (303) | High supply - Medium demand (503) |
| **High demand (5)** | Low supply - High demand (105) | Medium Supply - High Demand (305) | High supply - High demand (505) |

## 4. Results

### *4.1. Construction of the Supply Model of UPGS*

#### 4.1.1. Measurement of the Quality of UPGS

This study used Baidu's population heat value to represent UPGS visits. Firstly, the UPGS visits were given a Pearson correlation analysis with UPGS's quality factors GS, CC, WB, and PF (Table 4). Second, in the AHP, this study used the Pearson correlation coefficient q as a reference for the importance comparison among GS, CC, WB, and PF. According to the idea of the Saaty 1-9 scale method and the results of factor detection, the principle of constructing the comparison matrix was determined: if the q-value of Ci is 0-0.1 larger than that of Cj, then (Ci, Cj) = 1, where i, j = 1,2,3,4. Similarly, if the q-values of two factors differ by 0.1–0.2, 0.2–0.3, 0.3–0.4, the values filled in the comparison matrix will be 3,5,7 and 1/3,1/5,1/7. Based on the above principle, the pairwise comparison matrix was obtained (Table 5). GS, CC, WB, and PF represent GS, CC, WB, and PF, respectively. The CR is 0.1, and the weights are reasonably distributed. Based on the above calculation, the final weights of each factor of park quality were obtained as 0.216, 0.216, 0.054, and 0.514. Park quality was calculated as in Equation (6).

$$GQ = 0.216*GS + 0.216*CC + 0.054*WB + 0.514*PF \qquad (6)$$

**Table 4.** Pearson correlation analysis.

|  | GS | CC | WB | PF |
|---|---|---|---|---|
| Population heat value (Saturday) | 0.542 ** | 0.527 ** | 0.311 ** | 0.673 ** |
| Population heat value (Friday) | 0.697 ** | 0.658 ** | 0.513 ** | 0.721 ** |
| Population heat difference | 0.529 ** | 0.515 ** | 0.296 ** | 0.667 ** |

** indicates significant correlation at the 0.01 level (two-tailed).

**Table 5.** Factor pairwise comparison matrix.

|  | C1 | C2 | C3 | C4 |
|---|---|---|---|---|
| C1 | 1 | 1 | 5 | 1/3 |
| C2 | 1 | 1 | 5 | 1/3 |
| C3 | 1/5 | 1/5 | 1 | 1/7 |
| C4 | 3 | 3 | 7 | 1 |

$\lambda_{max}$ = 4.0735; CI = 0.0245; CR = 0.0275 < 0.1.

#### 4.1.2. Combine UPGS's Service Capacity and UPGS's Quality to Build a Supply Model of UPGS

As in Equation (7), TPSi is the sum of the supply of UPGS of different types within sub-district i. SPSi is the supply of small park within sub-district i, CMPSi is the supply of community parks within sub-district i, RPSi is the supply of regional parks within sub-district i, and CPSi is the supply of city parks within sub-district i. i is the sub-district number (i = 1, 2, ... 133).

SPSi, CMPSi, RPSi, and CPSi were calculated as in Table 6. a is the small park, b is the community park, c is the regional park, and d is the city park. Sai is the service area of the small park in sub-district i, Sbi is the service area of the community park in sub-district i, Sci is the service area of the regional park in sub-district i, and Sdi is the service area of the city park in sub-district i. PQa is the quality of small park, PQb is the quality of community park, PQc is the quality of regional park, PQd is the quality of city park, and SR is the service radius. Figure 3 shows an example of calculating the total supply of UPGS within the Sijiqing sub-district.

$$TPSi = SPSi + CMPSi + RPSi + CPSi \qquad (7)$$

**Table 6.** A formula for calculating the supply of UPGS within a sub-district.

| Calculation Formula | The Number of UPGS | Service Radius |
|---|---|---|
| $SPSi = \sum Sai * PQa$ | a = 1, 2, 3 …. n1 (n1 = 57) | SR = 500 m |
| $CMPSi = \sum Sbi * PQb$ | b = 1, 2, 3 …. n2 (n2 = 150) | SR = 1000 m |
| $RPSi = \sum Sci * PQc$ | c = 1, 2, 3 …. n3 (n3 = 48) | SR = 2000 m |
| $CPSi = \sum Sdi * PQd$ | d = 1, 2, 3 …. n4 (n3 = 57) | SR = 3000 m |

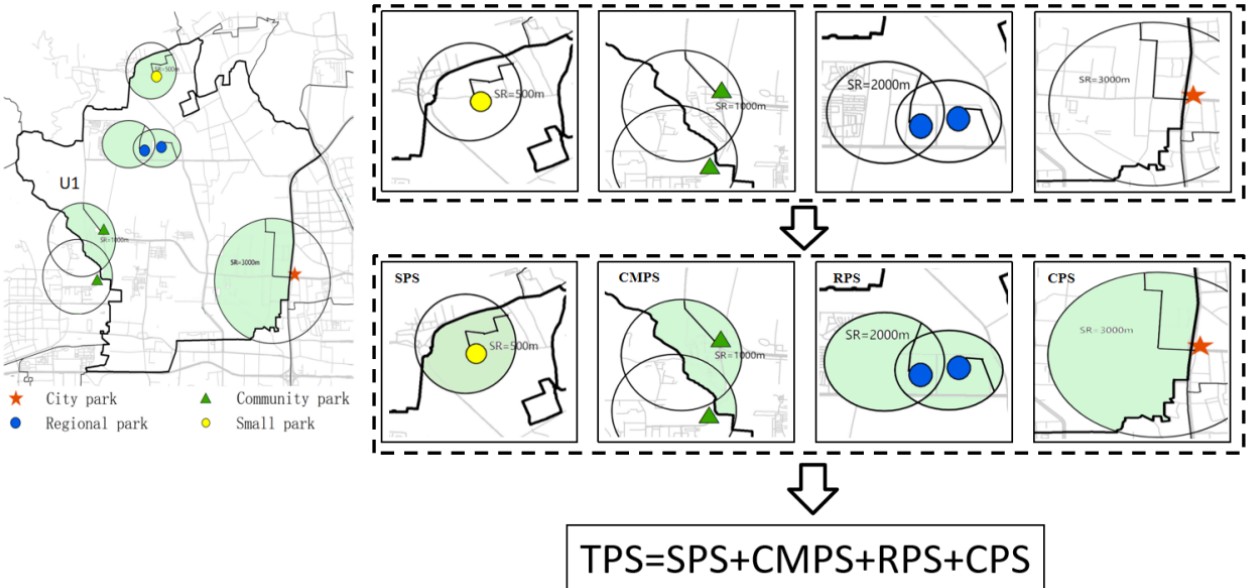

**Figure 3.** An example of the calculation of the total supply of UPGS within the Sijiqing sub-district.

### 4.2. Spatial Analysis of Supply of UPGS

There is an obvious spatial heterogeneity in the total supply of UPGS (TPS) enjoyed by the sub-district in the study area (Figure 4, left). In general, the spatial distribution of TPS shows a trend of first increasing and then decreasing from the urban center to the periphery and is characterized by higher TPS in the southwest and lower TPS in the northeast. Specifically, a few sub-districts (5) enjoy high-supply (Table 7), 25.0% of which is distributed in Chaoyang District, such as the Olympic Village sub-district; 25.0% in Fengtai District, such as the Lugouqiao and Huaxiang sub-district; and 20.0% in Haidian District, such as the Sijiqing sub-district, which has the highest TPS in the whole study area. These sub-districts are densely packed with parks, including the Olympic Forest Park, Capital Pear Garden, and Yue Gezhuang Garden (Figure 4, right), and these UPGS are of high quality and service capacity. The sub-districts with medium-supply (39) are clustered between the 4th and 5th rings in the eastern part of the study area and outside the 3rd ring in the west, such as Asian Games Village, Tiantan, Shichahai sub-district, etc., which are characterized by overall concentration and local dispersion. The reason is that the city parks and regional parks are concentrated in the area; these city parks and regional parks have strong service capacity and high quality. The sub-districts with low supply (89) are clustered in the central and eastern parts of the study area, such as Andingmen, College Road, Qinghua Park, and Zhongguancun sub-district. This is because many small parks are clustered in the central part of the study area, and such parks have low service capacity and park quality. The eastern part of the study area has less park green space, so both the central and eastern parts of the study area have a low supply of UPGS.

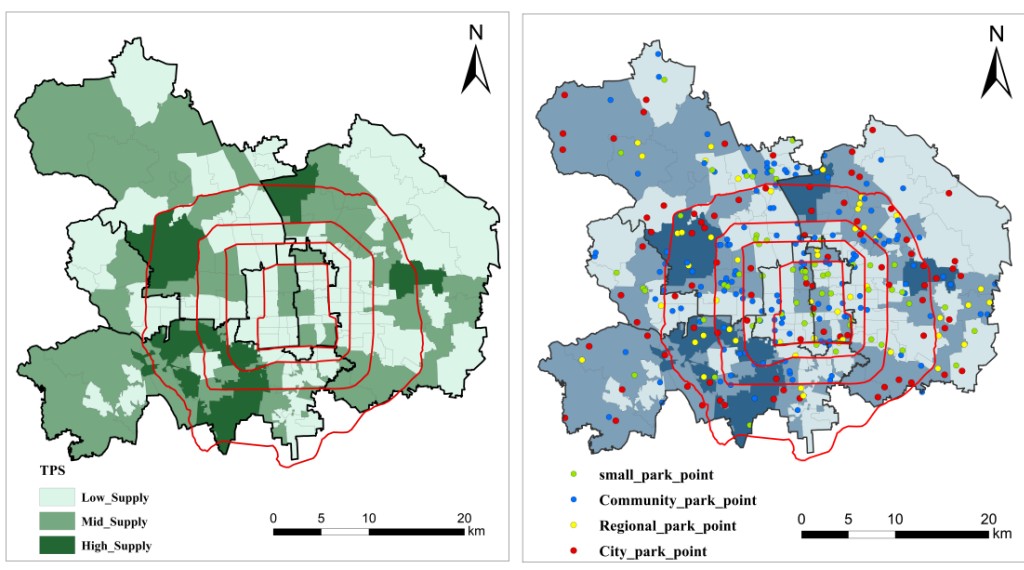

**Figure 4.** Spatial pattern of TPS (**left**); spatial pattern of UPGS (**right**).

**Table 7.** Statistics of TPS.

|              | Study Area (133) | Dongcheng | Xicheng | Chaoyang | Haidian | Fengtai | Shijingshan |
|--------------|------------------|-----------|---------|----------|---------|---------|-------------|
| Low-Supply   | 89               | 13        | 13      | 26       | 18      | 12      | 7           |
| Mid-Supply   | 39               | 4         | 2       | 14       | 10      | 7       | 2           |
| High-Supply  | 5                | 0         | 0       | 2        | 1       | 2       | 0           |

### 4.3. Fairness Evaluation of TPS

Figure 5 (left) shows the Gini coefficients for the ground-average TPS in the study area and the six districts, avoiding the effect of differences in geographic area between sub-districts on the Gini coefficients. The ground-average TPS is the TPS within each sub-district divided by the area of each sub-district. The Gini coefficient for the study area is 0.387, and the Gini coefficients within the six districts are: Fengtai (0.386) > Haidian (0.382) > Chaoyang (0.352) > Dongcheng (0.276) > Xicheng (0.265) > Shijingshan (0.261). Therefore, the current distribution of TPS in the central urban district of Beijing is relatively reasonable, without considering the residents' demand. It is worth noting that, according to the golden mean, the Gini coefficient of Fengtai and Haidian districts exceeds 0.382, reaching the alert line of green space resource distribution, which should draw the attention of the government and society.

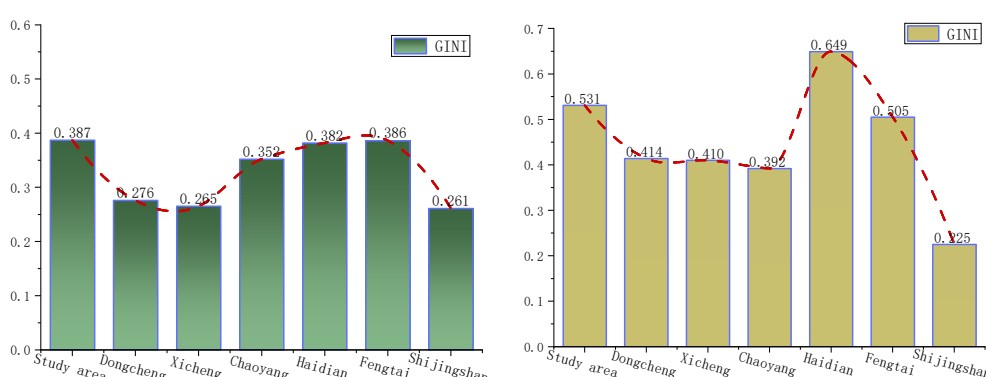

**Figure 5.** Gini coefficient of ground-average TPS (**left**) and Gini coefficient of per capita TPS (**right**).

Figure 5 (right) shows the Gini coefficient for the per capita TPS enjoyed within the sub-district of the study area and the six urban districts. The per capita TPS in a sub-district

is the TPS within each sub-district divided by the number of residents in each sub-district. The Gini coefficient for the study area is 0.531, indicating a highly unfair distribution of TPS when considering the residents' demand. In other words, residents do not have fair access to the recreational and related health benefits of UPGS. The Gini coefficients within the six urban areas are: Haidian (0.649) > Fengtai (0.505) > Dongcheng (0.414) > Xicheng (0.410) > Chaoyang (0.392) > Shijingshan (0.225). The results show that, except for Shijingshan District, the other five districts show different degrees of unfair distribution of TPS. Among them, Haidian District has the most uneven distribution of per capita TPS, followed by Fengtai, Dongcheng, Xicheng, and Chaoyang. In other words, compared to Shijingshan District, residents living in Dongcheng, Xicheng, Chaoyang, Haidian, and Fengtai are more likely to feel unfairness in the TPS, with the strongest sense of unfairness among residents living in Haidian District.

### 4.4. The Spatial Matching Relationship between the TPS and Residents' Demand

The above study used the Gini coefficient to investigate the fairness of the distribution of ground-average TPS and per capita TPS in the sub-district of Beijing. It is worth noting that the Gini coefficient only expresses the fairness of the distribution of green space resources as a whole and cannot show the spatial match between the distribution of public green space resources and the distribution of the resident population [52]. Therefore, this study further explored the spatial matching relationship between the TPS and residents' demand to find out the sub-district with insufficient TPS. The study's results can provide a reference for the refined management of UPGS.

#### 4.4.1. Spatial Matching of the TPS within the Sub-District to the Total Population of the Sub-District

The total population of the sub-districts shows a similar spatial distribution pattern as the TPS in the sub-districts (Figure 6 left). From the central area to the edges, the number of residents in the sub-district first increases and then decreases. Sub-districts with low demand are clustered in the Dongcheng and Xicheng districts within the Second Ring Road and the periphery of the study area, such as the Chaoyangmen, Sanlitun, Xiangshan, Changxindian, and Wulituo sub-districts. The East and West urban areas are the old districts of Beijing with a small geographic area; they are the cultural heritage gathering places of Beijing, with more cultural space and less residential space, so this is a low-demand area. The periphery of the study area, though large in geographical area, is mostly mountainous space with little urban space, so this is also a low-demand area. The sub-districts with medium demand are gathered and distributed between the second and fifth rings and the eastern part outside the fifth ring, such as Zhongguancun, Olympic Village, and Sanjianfang sub-district. This area covers most of the sub-districts in Haidian, Chaoyang, and Fengtai Districts. These sub-districts have experienced rapid economic development in recent years, attracting many migrant workers. Sub-districts with high demand are fewer and are distributed near the second to fifth rings, such as College Road, Xisanqi, Qinghe, and Xibeiwang sub-districts. Many internet companies, research institutes, and universities are located within the area, such as Xiaomi, Baidu, Qinghua, and Peking University. The opportunity for development attracts many people, so the area has a large population.

From the spatial matching analysis of the TPS within the sub-district and the residents' demand in the sub-district (Figure 6 right), 52.6% of the sub-districts in the study area enjoy a balance between the TPS and the demand of the residents in the sub-district (Figure 8). This area is located in the central area of the study area and the northwest and southeast areas, such as Shichahai, Sanlitun, Zizhuyuan, Xiaohongmen, Sujiatuo sub-district, etc. Since the TPS in the central part of the study area is low and the number of residents is low, the TPS can meet the demand of residents; in the northwest and southeast areas of the study area, the TPS is higher, so the TPS and the demand of residents can also reach a balance. However, the TPS in 33.8% of the sub-districts in the study area cannot meet the residents' demand. Among them, sub-districts with low-high should draw the most

attention, such as Xisanqi, College Road, and Fengtai sub-district, etc. The government should increase the TPS in this area by building small parks and other measures. The gap between supply and demand is smaller for sub-districts with low and medium traffic, such as Qinghe and Zhongguancun sub-district, etc. The government should increase the TPS in this area through indirect measures such as improving the road network and enriching the internal facilities of UPGS. TPS exceeds resident demand in 13.5% of the sub-districts in the study area, including the Olympic Village, Gucheng, and Pingfang. This oversupply is mainly due to the presence of many high-quality UPGS in the sub-district and the relatively low number of residents in the sub-district compared to the surrounding sub-district.

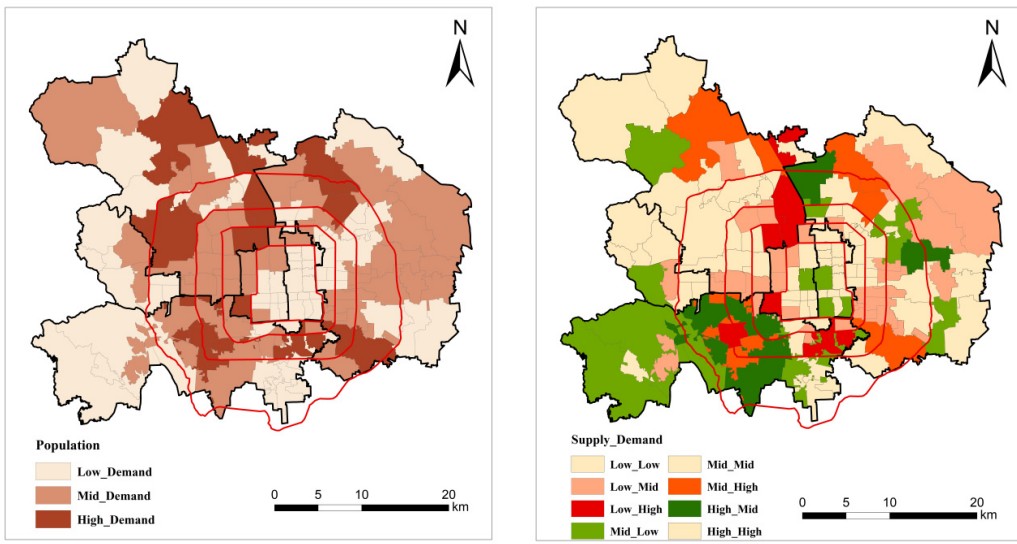

**Figure 6.** The total population' spatial distribution pattern in sub-districts (**left**); Spatial matching of the sub-district's TPS to the demand of total population within the sub-district (**right**).

To sum up, exploring the matching relationship between TPS and residents' demand clarified that more than one-third of the residents live in the under-supplied sub-districts in the study area that do not enjoy sufficient TPS. In order to improve the fairness of UPGS in the central urban district of Beijing, the government should take corresponding measures to increase its TPS in the under-supplied sub-district, such as improving the road network, increasing public service facilities in the UPGS, and creating habitats in the UPGS by planting.

4.4.2. Spatial Matching of the Sub-district's TPS to the Demand of Different Social Groups within the Sub-District

The theory of spatial justice emphasizes that the fairness of TPS should not only consider spatial fairness, but also social fairness [53–55]. Therefore, to complement the existing social justice content, we further explore the spatial matching relationship between TPS and the needs of different social groups.

The elderly, teenagers, and females all show similar spatial distribution characteristics to the total population (Figure 7a–c), and the number of people within the sub-district increases first and then decreases from the central area of the study area to the periphery. The reason is that the population base is small in the central and peripheral areas of the study area, so the numbers of elderly people, teenagers, and females are all smaller, but the distribution characteristics of the three groups differ spatially. In the case of the elderly, the distribution of sub-districts with high demand is relatively concentrated compared to the teenage and female groups, clustered between the second and fifth rings, such as Bailizhuang, Zhongguancun, and Datunlu sub-districts. This is due to the large population base between the second and fifth rings. In terms of teenagers, the spatial pattern of sub-districts with high demand is dispersed and distributed in the northern and southern parts of the study area, such as Xibeiwang, Xisanqi, Qinghe, and Shilidian sub-districts in

Haidian District. As far as the female group is concerned, sub-districts with high demand are clustered southwest and south of the Second Ring Road, among which most of them are distributed in Haidian and Fengtai districts, such as College Road, Qinghe, Xisanqi, and Xibeiwang sub-districts in Haidian District, and Dahongmen and Xincun sub-districts in Fengtai District.

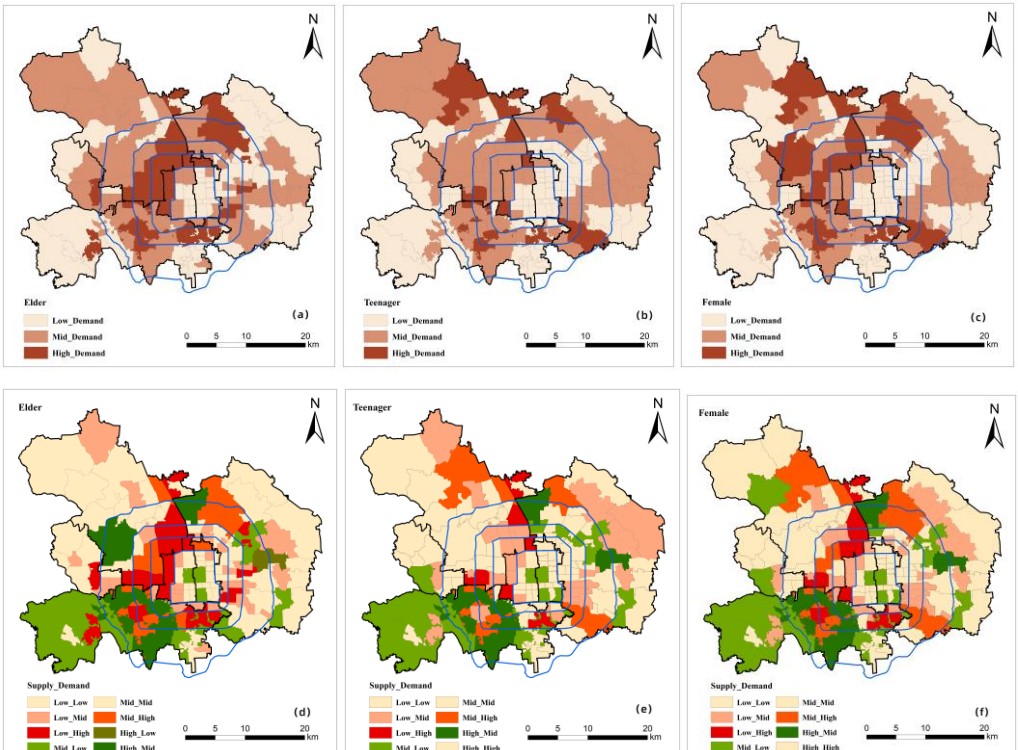

**Figure 7.** (**a**) Spatial distribution pattern of the number of elder in the sub-district; (**b**) Spatial distribution pattern of the number of teenagers in the sub-district; (**c**) Spatial distribution pattern of the number of females in the sub-district; (**d**) Matching the supply and demand of TPS with the elder; (**e**) Matching the supply and demand of TPS with the teenager; (**f**) Matching the supply and demand of TPS with the female.

In terms of spatial matching analysis between the TPS and residents' demand, 41.4% of the sub-districts in the study area have the TPS that matches the demand for the elderly within the sub-district (Figure 8). Insufficient supply exists on 46.6% of the sub-districts in the study area. These areas were distributed between the second and fifth rings. Most of the sub-districts with low-high are clustered in Haidian and Xicheng District, such as Beixiaoguan, Beitaipingzhuang, Wanshoulu, Guang'anmenwai, and Yuetan sub-district; a small number of sub-districts are distributed in Fengtai and Chaoyang District, such as Fengtai, Dahongmen, Jiuxiao, and Panjiayuan sub-districts. Compared with the spatial matching situation between the demands of teenagers and women and the TPS, the spatial matching between the elderly and the TPS is even less satisfactory. Among them, there are more sub-districts with Low-High (the number of residents' demand for the elderly), indicating that the elderly in the study area is disadvantaged in terms of access to TPS. A total of 12.0% of the sub-districts in the study area enjoy TPS that can adequately meet the needs of the elderly, and most of this area is distributed in Fengtai District, such as Changxindian, Wangzuo, and Huaxiang district, etc. A small portion is distributed in Chaoyang and Haidian District, such as the Olympic Village, Jiangtai, and Shiziqing sub-districts, etc. The sub-districts with high-low are distributed in Pingfang sub-district in Chaoyang District. It shows that the TPS in these areas far exceeds the demand of the elderly. It is worth noting that there are fewer sub-districts where the TPS adequately meets the needs of the elderly compared to the teenagers and female groups.

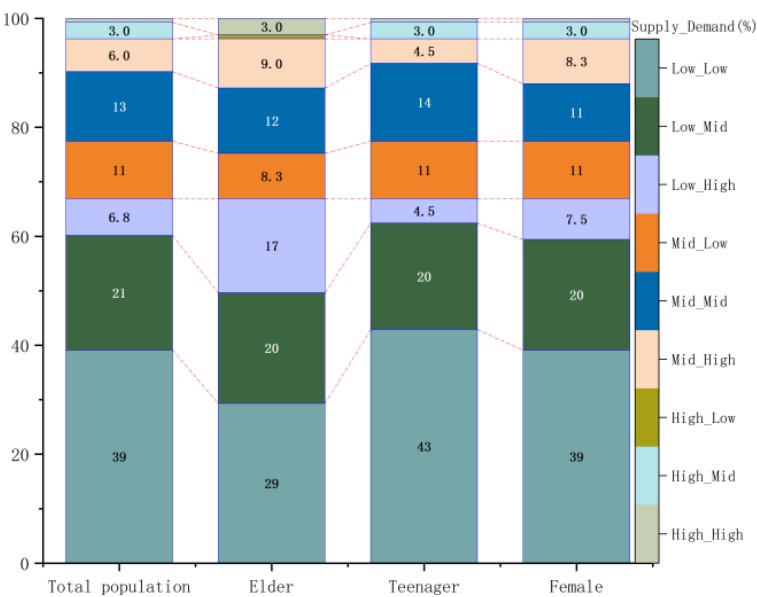

**Figure 8.** Statistics on the spatial match between TPS and residents' demand.

In terms of teenagers, 57.9% of the sub-districts in the study area have a TPS that matches the needs of youth (Figure 8); 13.5% of the sub-districts in the study area have a TPS that exceeds the needs of youth, and these areas are located in Changxindian and Wangzuo sub-districts in Fengtai District, Olympic Village and Pingfang sub-district in Chaoyang District, Donghuamen, Xichanganjie sub-district in Dongcheng and Xicheng District. It shows that more sub-districts in the study area enjoy a TPS that can meet the needs of teenagers compared to the female group. On the other hand, 28.6% of sub-districts in the study area do not have enough TPS to meet the needs of teenagers, mainly in Chaoyang, Haidian, and Fengtai Districts. Among them, 4.5% of the study area is low-high, and these areas are distributed in Haidian District, Xicheng District, and Fengtai District, such as Xisanqi, College Road, Wanshou Road, Desheng, Guang'anmenwai, and Dahongmen sub-district. For the sake of teenagers' healthy physical and mental growth, the TPS in this area should be improved as soon as possible.

As far as the female group is concerned, 50.4% of the sub-districts in the study area have a TPS that matches the female needs of the sub-district (Figure 8). Most of these areas located in the periphery of the study area, such as Sujiatuo and Jinzhan sub-districts. The TPS of 36.1% of the sub-districts in the study area is insufficient, with sub-districts with low-high accounting for 7.5% of the study area. This area is distributed in Haidian and Fengtai Districts, such as College Road, Xisanqi, and Fengtai sub-districts. It shows that although the TPS in Haidian District is high as a whole, it is difficult to meet the actual demand of women in Haidian District due to the relatively concentrated distribution of female groups and the large gap in the distribution of TPS. In the future, when optimizing UPGS in Haidian District, the government should focus on areas that cannot meet women's needs and improve their spatial quality. A total of 13.5% of the sub-districts in the study area enjoy a TPS that exceeds women's demand; the area with high-low is located in the Olympic Village and Pingfang sub-district in Chaoyang District and the Lugouqiao sub-district in Fengtai District.

Overall, 71.4% of the sub-districts in the study area meet the needs of teenagers, 63.9% meet the needs of females, and only 53.4% meet the needs of the elderly. This indicates that the elderly have less access to UPGS compared to the teenage and female groups. In summary, there are obvious differences in the spatial matching between the TPS and the demand of different social groups, and the overall level of matching between supply and demand is low, and many sub-districts are still in a state of undersupply of TPS. Among them, insufficient TPS for the needs of the elderly is the most obvious.

## 5. Discussion

### 5.1. The Validity of the Improved Supply Model of UPGS

This study introduced the Baidu heat map to improve the supply model of UPGS by integrating the quantity of UPGS, the service capacity of UPGS, and the quality of UPGS. The results show that the TPS is higher in Haidian and Chaoyang Districts and lower in Dongcheng and Xicheng Districts. This is because Haidian District is a construction area of "three mountains and five gardens", and there are famous parks such as Xiangshan Park, Beijing Botanical Garden, etc. For example, the red leaves in Xiangshan Park are a world-famous landscape that attracts people year-round and has a strong supply capacity. Chaoyang District covers many parks, such as Olympic Forest Park and Chaoyang Park, which are of high quality, often accommodating an average daily flow of 20,000 people during holidays [56]. On the other hand, Dongcheng and Xicheng District are cultural heritage gathering places, distributed in the center of Beijing, and bear the cultural functions of Beijing. Moreover, their geographical area is small, mostly dotted with small parks. The results of this analysis reflect the rationality of the supply model of UPGS.

### 5.2. Reasons for the Unfairness of the UPGS

This paper confirms extensive evidence that UPGS are unfairly distributed across social groups, supporting environmental justice theory. The study found inconsistent access to UPGS for the elderly, teenagers, and female groups within the central urban district of Beijing. Nearly 50% of the sub-districts in the study area do not meet the needs of the elderly. This indicates that the elderly have highly unfair access to UPGS in the study area, a finding that is consistent with previous studies [28]. From 2010 to 2020, the elderly population in Beijing increased by 2.709 million. The elderly are more concentrated in Chaoyang, Haidian, and Fengtai Districts [57], indicating that the aging phenomenon has become more serious in recent years, and the elderly population within sub-districts has increased. In addition, most of the elderly in Beijing are indigenous people, who cannot relocate. The UPGS is growing slowly, so most sub-districts cannot meet the needs of the elderly. A total of 71.4% of the sub-districts with TPS in the study area can meet the needs of teenagers. This indicates that teenagers prefer to live in a sub-district with a higher TPS, and this finding is consistent with He et al. For the healthy growth of teenagers, parents usually choose a sub-district with more TPS to live in. In total, 63.9% of the sub-districts with TPS in the study area can meet the needs of female groups, such as the Olympic Village, Pingfang, and Sujiatuo sub-districts, which have many city parks distributed in these sub-districts, which is consistent with previous studies [58], indicating that the female group prefers natural scenes or large-scale integrated parks.

### 5.3. Enhancement Strategies for Fairness of UPGS

According to the results of this study, there is obvious unfairness within the central urban district of Beijing. There are many sub-districts with insufficient supply. Therefore, to improve the fairness of UPGS, this paper makes some suggestions for sub-districts with insufficient TPS. This study measures the TPS based on the quantity of UPGS, the service capacity of UPGS, and the quality of UPGS, so the TPS can be increased in the following three ways: Additional small parks can be built in sub-districts with a large gap between supply and demand, such as Xisanqi and Xueyuanlu sub-districts in Haidian District. The service capacity of UPGS is affected by the scale of UPGS and road networks. With the same scale of UPGS, the road networks can be improved to increase the service capacity of UPGS, thus increasing the supply of UPGS. This study evaluates the quality of UPGS using the scale of UPGS, the area of water bodies in UPGS, the area of green areas in UPGS, and the number of public service facilities in UPGS. Therefore, the quality of UPGS can be increased by improving the number of public service facilities, thus increasing the TPS. Especially in the sub-district where the supply of elderly and teenagers is insufficient, we increase the facilities for elderly and teenagers' activities in UPGS to improve the fairness of UPGS.

### 5.4. Contribution

Sub-district units are closer to the actual living units of residents, and it is an important reference for urban refinement management to use them as the research scale in this study [59]. Many scholars study the spatial fairness of UPGS by choosing low-resolution green spaces in land cover data, ignoring green spaces with smaller dimensions, such as community parks and private gardens [60]. Based on this situation, this paper improved the supply metric model of UPGS, fully considers the UPGS of different types, such as small parks and community parks, and then introduced the Baidu heat map to construct the quality evaluation model of UPGS, which strongly avoids the subjectivity of determining the weights of quality factors of UPGS in the past. This method is objective, easy to operate, and universal. In addition, this study corrected the data of different social groups in the sub-district in 2020 based on the sixth population census data. This study used these data to measure the matching relationship between the TPS and residents' demand in Beijing, which can provide real-time guidance for planning in Beijing. Currently, studies on the spatial relationship between the TPS and residents' demand usually focus only on the excess and shortage of quantity, and deeper information mining is still insufficient [61]. The evaluation system proposed in this study more accurately measures the supply–demand matching relationship of UPGS. Compared with existing studies [62,63], the results of this study can better identify areas within the study area that are undersupplied, effectively guiding the optimal layout of UPGS in the study area.

### 5.5. Limitations and Future Research

Large-scale studies are more extensive, and the results will provide a more macroscopic understanding of the supply–demand match between the supply of UPGS and residents' demand, which will help address the problem of insufficient green space supply at the regional, provincial, and national levels [64]. Large-scale data collection is efficient and uniform, offering the possibility of measuring park green space supply on a large scale. This study aims to construct a multi-source data-based green space equity evaluation system to provide a paradigm for other cities. This study mainly explores the match between the supply of UPGS and residents' potential demand at the macro level to guide urban green space planning. However, the limitations of this study should be noted.

First, due to the limitation of data availability, only four factors, such as the scale of UPGS, the scale of water bodies in UPGS, the scale of green spaces in UPGS, and the number of services in UPGS, are selected as quality factors of UPGS in this study. More factors can be selected through field surveys, such as the number of seats inside the UPGS, the number of trash cans, the number of boulevards, etc.

Second, since 340 parks and 133 sub-districts were selected for this study; the scope of the study is large, and it is difficult to collect the real demand data of residents through questionnaires, so this study uses the census data within a sub-district instead of the potential demand of residents. Demand quantification has been a challenge in supply–demand matching studies. Many attempts have been made by many scholars to address solutions to this problem, such as social surveys, socioeconomic data, and social media data [11,65,66]. Although it seems reasonable to focus on residents' real demand for UPGS through questionnaires, it is difficult to accurately describe their real demand through questionnaires when the study area is large and residents' preferences for parks are different and complex. The small sample size of social media data with geographic coordinates makes it difficult to cover the elderly and children and cannot fully and truly reflect residents' demand [15]. Therefore, in large-scale studies, census data are increasingly being considered representative of potential residents' demand due to their comprehensiveness and accuracy [67]. On the contrary, in small-scale studies, research data collection is more convenient; the real preferences of residents can be directly obtained through questionnaires, and the research results can more accurately identify spatial mismatch areas and provide more accurate guidance for urban planning. In the future, we can focus on small-scale studies to explore the social equity of UPGS, analyze the acceptable threshold of residents'

travel distance through questionnaire surveys, and obtain more detailed data on residents' park use behavior and the use demand of each group through qualitative surveys [64].

Third, Beijing, as the capital of China, has relatively more parks, and most of them are open parks. This study selects 340 parks in the main urban area of Beijing that can provide people with leisure and recreation on a macro level, focusing on the internal quality of parks and ignoring the characteristics of the entrance fee system, opening hours, degree of openness of parks, etc. In the future, when studying the equity of green space in other areas, we can explore the openness of parks through field surveys and focus on the accessibility characteristics of parks.

## 6. Conclusions

This paper explores the fairness of UPGS in Beijing, aiming to enrich the empirical analysis of the fairness of UPGS in China. Firstly, this paper introduced Baidu population heat data and improved the supply metric model of UPGS by combining the quantity of UPGS, the service capacity of UPGS, and the quality of UPGS. Secondly, this study evaluated the fairness of UPGS from spatial and social fairness perspectives. Finally, this study explored the supply–demand matching relationship between the TPS and the total population and different social groups, respectively.

The improved supply model of UPGS clarifies the evaluation method of the quality of UPGS, measures the supply of UPGS of different types, and provides more detailed references for the refined management of Beijing.

The per capita supply of UPGS is unevenly distributed among the six urban districts of Beijing, especially in Haidian District, which may lead to a stronger sense of unfairness for residents living in Haidian District compared to those in the other five urban districts. There is a spatial mismatch between the TPS and the total population of the sub-district, and there are obvious differences between the supply and demand matching relationships between the TPS and different social groups. As a result, the overall supply and demand matching level is low, and many sub-districts are still undersupplied, among which the undersupply for the elderly is more obvious. This reflects the potential lack of TPS and the shortage of UPGS in the central urban district of Beijing.

**Author Contributions:** Conceptualization, X.W., Q.M. and X.L.; methodology, X.W. and Q.M.; software, X.W. and X.L.; validation, X.W., L.Z., M.A. and X.H.; formal analysis, X.W.; investigation, X.W. and Y.B.; resources, X.W.; data curation, X.W.; writing—original draft preparation, X.W. and X.H.; writing—review and editing, X.W., X.L., M.A., Y.B. and T.J.; visualization, X.W., X.L., M.A. and L.Z.; supervision, Y.B.; project administration, Q.M. and L.Z.; funding acquisition, Q.M. All authors have read and agreed to the published version of the manuscript.

**Funding:** This work was supported by the National Natural Science Foundation of China (grant number 42171357), the National Natural Science Foundation of China Major Program (grant number 42192580, 42192584), the National Natural Science Foundation of China (grant number 42201384)and the Bilateral Chinese-Hungarian Project (2019-2.1.11-TÉT-2020-00171).

**Data Availability Statement:** The data presented in this study are available on request from the author. The data are not publicly available due to privacy. Images employed for the study will be available online for readers.

**Conflicts of Interest:** The authors declare no conflict of interest.

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
