# Peer review of "Evaluation of Fairness of Urban Park Green Space Based on an Improved Supply Model of Green Space: A Case Study of Beijing Central City"

_remotesensing, doi:10.3390/rs15010244_

Round 1

Reviewer 1 Report

Thanks for inviting me to review this manuscript. The authors aimed to assess the spatial and social fairness of urban park green spaces based on the improved supply model provided by He et.al. Although the logic framework seems to be appropriate, there are indeed substantial issues in the present study, in relation to study design, research gap statement, methodology, and model selection that should be addressed.

Line 30, please simply specify the demand aspect.

Line 31-34, please provide more quantitive statements regarding spatial fairness(e.g., Gini=xx).

Line 69, How to evaluate the spatial fairness via the Gini coefficient and Lorenz curve without considering the number of urban residents?  

Line 83, To my best knowledge, many studies have considered the quality of UPGS in the ‘green injustice’ paradigm.

Line 93, why did the authors mention OSM here? Are they quality factor data of UGGS from big datasets?

Line 115, the authors have incorporated ‘different social groups” in their framework. I suggested that the authors should consider social justice simultaneously. As they mentioned, social justice is based on social fairness and emphasizes that the capabilities and needs of various groups are different.

Line 183, specify the data resolution of the population heat value via the Baidu heat map. Please justify the availability and usability of the heat map data. Will the low-resolution database yield bias in delineating the densities of the urban population?

Line 205, green space or green areas?

Line 273-284, these paragraphs should be clarified. How to evaluate the matching supply and demand? Did the authors consider the spatial autocorrelation of green space provisions?

Reviewer 2 Report

Evaluation of Fairness of Urban Park Green Space Based on an Improved Supply Model of Green Space: a Case Study of Beijing Central City Beijing parcs

This paper is very interesting because it highlights a burning issue related to the ratio of public green spaces in relation to city residents, especially at a time when they are particularly necessary for the better health of people. 

However, it should consider some more qualitative aspects of the issue and make the necessary corrections to the following:

Observation 1

p1. "This study aims to develop a fairness assessment model to quantify the fairness of UPGS distribution and the matching relationships between supply and demand local requirements or needs for UPGS. 

To achieve the aims of the study, we improved the supply model of UPGS by integrating three factors:  the number of UPGS, the service capacity of UPGS, and the quality of UPGS in the Beijing central city. Subsequently, we evaluated the spatial fairness and social fairness of the supply of UPGS using the Gini coefficient. Then we quantitatively analyzed the spatial matching relationship between the supply of UPGS and residents' demandlocal requirements or needs" 

In this case it is better to say, "local requirements or needs" and not "demand"

"Local", because the counts are only for locals’ residents and not for other visitors, coming from other areas,

"Needs", because the survey did not measure demand for the parks with questionnaires

Observation 2

p1. "The results show that (a) the improved supply model of UPGS can measure the supply of UPGS of different types in a more detailed way. (b) Residents in the Beijing central city do not have fair access to the supply of UPGS. While residents in Haidian District have the highest sense of unfairness. followed by Fengtai, Dongcheng, Xicheng, and Chaoyang District. (c) The matching relationship between the supply of UPGS and the needs of different social groups is not ideal, especially the spatial matching relationship between the needs of the elderly and the supply of UPGS".

p.18 : "residents live in the Haidian District have a stronger sense of unfairness about the distribution of TPS than residents in five urban areas".

The above evaluation requires qualitative research with interviews and questionnaires in the parks themselves, which has not been done and therefore cannot be based on data alone. It should be reformulated. e.g. The data shows an uneven distribution of facilities in the designated parks, which may create a sense of injustice.

Observation 3

The model should also consider whether admission to the parks is free, or visitors must pay. This can produce a stronger sense of unfairness. 

You should also check whether there is a lot of pressure from their touristification. 

You should also look at accessibility (accessibility to metro stations, bus stations etc.), access for disabled people, benches, kiosks etc.

Observation 4 

It would be a good idea to consider private parks as well. In this case the inequalities in access to parks will turn out to be much greater. 

Observation 5 

You should also define how you used remote sensing to identify parks. By photogrammetry and combining it with cartographic data? Did you take vegetation indicators into account to identify areas that have greater green cover and therefore greater need for green public spaces? 

Observation 6 

Finally, the abandonment of some historic traditional parks may be related to efforts to degrade the surrounding area in order to lower the values of residential property values and expropriate them by larger real estate companies in the direction of so-called green gentrification.

Beijing's historical parks have a special life throughout the day and night, and this is something that should be positively evaluated and enhanced, taking into account, for their maintenance, the opinion of the users themselves.  

Reviewer 3 Report

General comments

The paper entitled “Evaluation of Fairness of Urban Park Green Space Based on an Improved Supply Model of Green Space: a Case Study of Beijing Central City” stresses the important role in providing ecological and social benefits of Urban park green space (UPGS) and results are useful in support for decision-making in optimizing UPGS and providing a reference for a careful urban management. The aim of the work is well explained in terms of research design and procedures; methodology described in a clear way and complete of details. The structure of the paper is well balanced for length of the sections and bibliography. English is fluent, concise in the concepts and thus clear.
